# Geochemical Conditions and Factors Controlling the Distribution of Major, Trace, and Rare Elements in Sul Hamed Granitic Rocks, Southeastern Desert, Egypt

Neveen S. Abed [1], Mohamed G. El Feky [1], Atef El-Taher [2], Ehab El Sayed Massoud [3,4], Mahmoud R. Khattab [1], Mohammed S. Alqahtani [5,6], El Sayed Yousef [2,7] and Mohamed Y. Hanfi [8,9,*]

1   Department of Geochemical Exploration, Nuclear Materials Authority, 530 El Maadi, Cairo 11843, Egypt
2   Department of Physics, Faculty of Science, Al-Azhar University, Assuit 71452, Egypt
3   Department Biology, Faculty of Sciences and Arts in Dahran Aljanoub, King Khalid University, Abha 61421, Saudi Arabia
4   Agriculture Research Center, Soil, Water and Environment Research Institute, Giza 12112, Egypt
5   Department of Radiological Sciences, College of Applied Medical Sciences, King Khalid University, Abha 61421, Saudi Arabia
6   BioImaging Unit, Space Research Centre, Department of Physics and Astronomy, University of Leicester, Leicester LE1 7RH, UK
7   Physics Department, Faculty of Science, King Khalid University, Abha 61413, Saudi Arabia
8   Department of Medical and Radioactive Researchers, Nuclear Materials Authority, 530 El Maadi, Cairo 11843, Egypt
9   Institute of Physics and Technology, Ural Federal University, St. Mira, 19, 620002 Yekaterinburg, Russia
*   Correspondence: mokhamed.khanfi@urfu.ru

**Abstract:** Egypt is mainly covered by ophiolitic rocks, muscovite, and two mica granites, in addition to different types of acidic and basic dikes. Our field observations indicated that El Sela granites were subjected to alteration types such as silicification, kaolinization, and hematitization, which is associated with uranium mineralization. Petrographic investigations clarified that these rocks were affected by saussiritization, muscovitization, and silicifications as the main alteration types associated with uranium mineralization (uranophane and autunite). We carried out chemical analyses of our samples for major oxides and trace and rare earth elements using ICP-OES and ICP-MS. The studied altered granites had high silica, titanium, and phosphorous as major components, with enhanced amounts of trace elements such as Nb, Ta, Zn, Mo, Pb, and Re, in addition to REE, especially light ones. The average REE content was higher than that of worldwide granites with LREE enrichment. One sample had a strong M-type tetrad effect in the fourth type; other samples had weak W-type in the third type, indicating the effect of hydrothermal alteration processes in the altered granites. This was confirmed by calculating the ratios of most isovalents that deviated from the chondritic ratio in many values. Variation diagrams of U and some trace elements illustrated that U had a weak positive correlation with Y and a strong positive correlation with gold, while it had weak to moderate negative correlation with Hf and Zr/U. In addition, uranium had a weakly defined correlation with the other trace elements, indicating a weak to moderate effect of magmatic processes, while the post-magmatic processes surficial or underground water greatly influenced the redistribution of uranium and other elements.

**Keywords:** geochemical conditions; Sul Hamed; granitic rocks; rare and rarest elements; tetrad effect

## 1. Introduction

Geochemistry is a direct method for detecting anomalies that measures the actual indicator elements or compounds that are being sought. Indicator or pathfinder elements are minerals or elements found near the elements being sought, and they are easier to discover or have more visible aberrant characteristics than the main focus of the search [1]. Because

of the dissolution or replacement of the primary components and accessory minerals, as well as the development of new mineral phases, practically all of the major oxides and trace elements are mobilized during a hydrothermal alteration.

Deuteric and post-magmatic hydrothermal events have an impact on granites. Hematitization, sericitization, episyenitization (desilicification), and fluoritization are hydrothermal modifications linked with U-mineralization in the host granites. Microclinization, sericitization, and muscovitization of alkali feldspar and plagioclase are all signs of alkali metasomatism. The uncommon Zr/Hf, Sr/Eu, Nb/Ta, La/Nb, La/Ta, Rb/Sr, and U/Th ratios indicate hydrothermal alteration during the late magmatic stage. The highly developed nature of these rocks is also reflected in their uneven REE pattern, which is especially obvious in late magmatic phases with substantial hydrothermal or deuteric alteration, as seen by K-metasomatic and fluoritization [2].

Rare earth elements (REE) are mostly found in accessory minerals in igneous and metamorphic rocks, and they have little mobility in these high-temperature environments [3]. Conversely, hydrothermal and weathering fluids can mobilize REE primarily through the formation of soluble complex ions [2]. The concentration of REE in soils is dependent on a variety of factors, including the composition of the source rocks, weathering processes, and organic matter content; but in general, light REE (LREE) predominate over heavy REE (HREE) in the shallower layers [4]. Zhao et al., 2008 [5] suggest that the new MW-type of tetrad effect is likely to be caused mainly by the interaction of aqueous liquids with alkaline rocks. Mahdy and El Kammar, 2003 [6] in Kab Amiri granite from CED, Egypt, revised the convex (M-type) $T_1$ that accompanies the concave (W-type) $T_3$ to the physico–chemical conditions that prevailed during the alkali-metasomatism of the Kab Amiri granitoids, suggesting severe changes in the physico–chemical conditions in the box-cut granites. El Mezayen et al., 2019 mentioned that the peculiar MW-type tetrad effect might be an indicator for the Au mineralization of reworked plutons [7].

In chlorides, nitrates, and perchlorates, the REE are more soluble than they are in carbonates, hydroxides, and fluorides) [8]. The chemical behaviour of REE is governed by their ionic radii, charge, volatility, and the stability of their complexes, according to McLennan, 1994 [9]. The tetrad or double-double factor was defined as a significant increase in stability associated with the half (Gd) and fully filled (Lu) 4f electron shells, as well as smaller but still significant increases in stability associated with the quarter (between Nd and Pm) and three-quarters filled (between Ho and Er) 4f shells. The tetrad effect is thought to be most visible in natural waters or rocks that have come into contact with fluids [10].

REE distributions in geological samples are controlled by the tetrad behavior on the REE legends; hence, REE representing quarter, half, three-quarters, and fully filled 4f shells have higher stability [9]. In natural samples, two complimentary types of tetrad effects have been identified: REE in a solid phase exhibit an M-type tetrad effect, whereas REE in a solution exhibit a W-type tetrad effect [11]. REE-bearing rock-forming minerals, monazite and xenotime, may also display a W-type tetrad effect—rather weak but sufficient for causing an M-type tetrad effect in the remaining melt [12,13].

The occurrence of the tetrad effect in nature has been linked to a history of contact or interaction with water [14]. Processes such as high degrees of fractional crystallization, hydrothermal alteration, and mineralization are commonly associated with tetrad effects in REE patterns. In rare metal granites, tetrad behavior is observed [15,16].

With increasing ligand concentration, the tetrad effect becomes more noticeable. Since Y with no 4f electrons is a pseudo-lanthanide that behaves like Ho under aqueous conditions, the positive divergence of Y/Ho from the chondritic ratio might be a property of both the W-type and M-type tetrad effects [17]. If the M-type tetrad effect occurred from a favoured removal of Ho from the solid to liquid phase relative to its neighbors, Y is likely to be less soluble in water and preferentially housed in or adsorbed on the solid phase, in contrast to Ho.

Geochemically, two-mica granite boreholes are A-type granites and peraluminous characteristics. They are enriched in large ion lithophile elements (LILE; Ba, Rb, and Sr),

high field strength elements (Y, Zr, and Nb), and LREE but depleted in HREE with a negative Eu anomaly [18].

Our study aimed to determine the main factors controlling the distribution of major, trace, rare, and rarest elements (Au, Re, and In) in the studied, mineralized granites of the Sul Hamed area and investigate the correlation between uranium and other mineralization.

## 2. Material and Methods

### 2.1. Geological Settings

The study region is about 30 km southwest of Abu Ramad, a city in Egypt's Eastern Desert, and its boundaries are latitudes 22°13′–22°20′ N and longitudes 36°08′–36°20′ E. It is mostly covered by ophiolitic rocks; Sul Hamed, 599 m.a.s.l., has muscovite granites; Qash Amer, 724 m.a.s.l., has two mica granites; El Sela, 557 m.a.s.l., has acidic and basic dikes; and El Sela, 557 m.a.s.l., has ophiolitic rocks (Figure 1). The Sul Hamed ophiolite dips steeply to the SE. The main sequence of the Sul Hamed ophiolite, which is located in the western part of the study area, comprises three NE–SW trending subvertical lithological zones: (a) serpentinites and related rocks in the NW, (b) metagabbros in the central zone, and (c) metavolcanics (sheeted metabasalt) to the southeast of Sul Hamed ophiolite.

El Sela granitic pluton is composed of two-mica (muscovite–biotite) granite intrusions (Figure 1). It is jointed in different directions and defines low to moderate reliefs. This granite is injected with microgranite, dolerite and bostonite dikes, and quartz and jasperoid veins. They are mostly injected along E–NE-W–SW and/or N–NW-S–SE to N–S directions, which represent the most important tectonic trends for U anomalies in the study area. Pegmatite pockets are found as irregular bodies in two-mica granite. A microgranite dike hosts the most radioactive anomalies in the study area. It is injected into the two-mica granite along an E–NE/W–SW shear zone with a dip about 75° to the south. It is whitish pink, pale pink, and reddish pink to pale grey in color and is 3 to 20 m thick. The El Sella shear zone, trending E–NE/W–SW, dissects the northern segment of the El Sella muscovite–biotite granite, extends over a distance of 1.5 km, and varies in width from 1 to 35 m. The granite becomes mylonitized and cataclased within the shear zone. From the structural point of view, this shear zone is cut and displaced into three separated parts by two N–NW/S–SE-trending strike-slip faults. Uranium mineralization in the studied area is recorded in both El Sela sheared granite and its injected microgranite dyke. Ferrugination, silicification, and kaolinitization a with few dark patches of manganese dendrites are the main rock alterations that developed within the studied shear zone. Ferruginated and kaolinitized granites with a few dark patches of manganese dendrites are the main wall-rock alteration features that developed within the studied shear zone. These alterations are more pronounced on both sides of the shear zone and cut by quartz veins. Field evidence, textural relations, and the composition of ore minerals suggest that the main mineralizing event was magmatic (615 ± 7 Ma and 644 ± 7 Ma CHIME monazite), in the El Sella shear zone, with later hydrothermal alteration and local remobilization of the high-field-strength elements [19]. Scanning electron microscope (SEM) and electron microprobe analyses (EPMA) were used to identify the studied columbite, zircon, uranium, and thorium minerals (cheralite, uranothorite, and huttonite monazite), along with the zircon (Hf), monazite, xenotime, rutile, pyrite, lepidolite, phlogopite, and fluorophlogopite in the El Sella shear zone [19–21].

### 2.2. Petrography

El Sella granite is medium-grained monzogranite, with a grain size ranging from 3.0 to 3.5 mm in length. It is composed of plagioclase (32.5% of the rock), alkali feldspars (26.7% of the rock), and quartz (37.4% of the rock) as essential minerals associated with mica minerals (3.4% of the rock).

Plagioclase is the main feldspar found as euhedral crystals of oligoclase ($An_{16}$), characterized by lamellar twining. Occasionally, some crystals are fractured and saussiritized, especially around the shear zones (Figure 2a). Alkali feldspars present as euhedral crystals

of perthite and antiperthite (Figure 2b). Quartz is found as equant crystals with wavy extinction. The mica minerals are represented by muscovite and biotite as primary flakes corroded by the feldspars and as secondary flakes of muscovite after biotite, which is characterized by moderate pleochroism (Figure 2c).

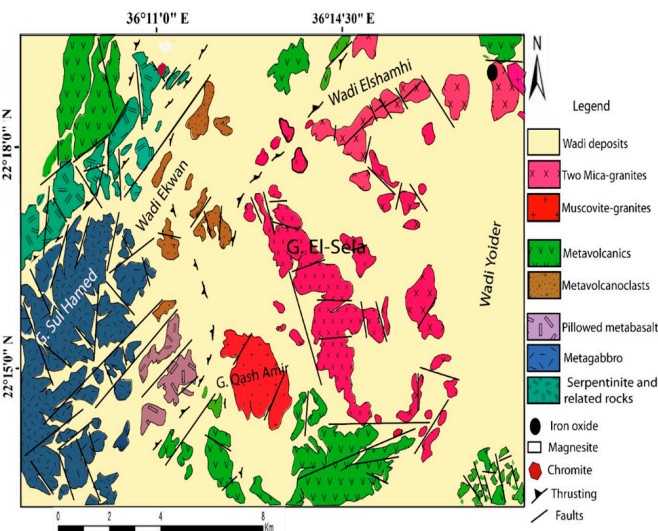

**Figure 1.** Geologic map of El Sella area [22].

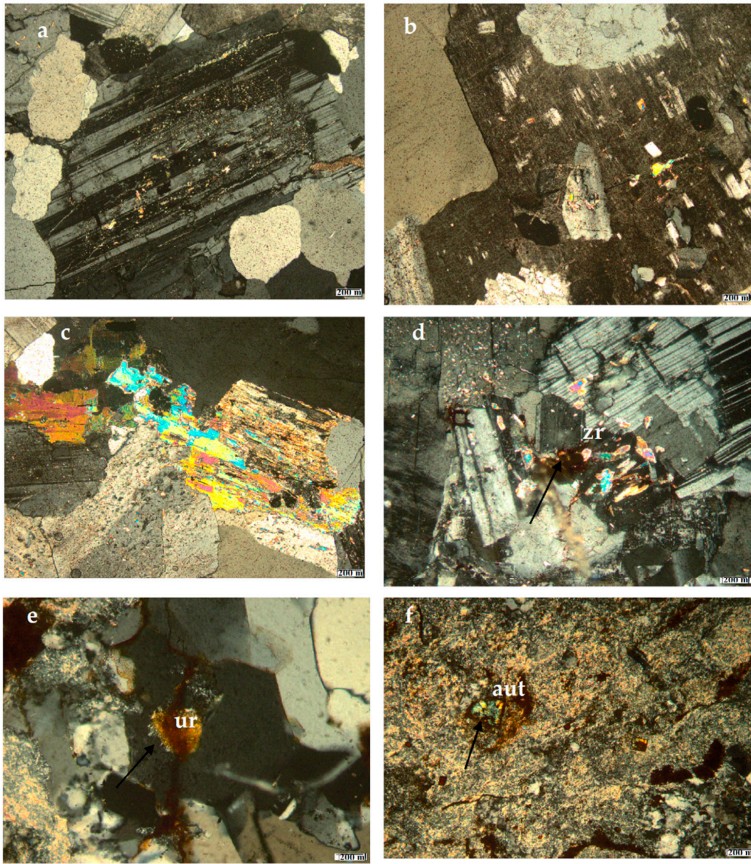

**Figure 2.** (**a**) Cracked crystal of plagioclase with partial saussiritization. (**b**) Euhedral crystal of antiperthite. (**c**) Flakes of biotite partially transformed to secondary muscovite, associated with plagioclase. (**d**) Well-formed crystals of zircon coated by iron oxides, associated with plagioclase. (**e**) Radioactive mineral that may be uranophane in quartz. (**f**) Well-formed crystals of autunite, associated with silicification.

The accessory mineral is mainly zircon, in addition to the radioactive minerals that are represented by uranophane and autunite. Zircon is faintly metamictized and fractured by effect of the radioactive damage (Figure 2d). Uranophane is present, associated with quartz and related to the secondary silica (Figure 2e). Autunite occurs as tabular crystals with rectangular outlines characterized by low relief and a lemon-yellow color (Figure 2f). Conversely, few crystals of violet fluorite were recorded in the samples that were picked from the shearing zone.

### 2.3. Analytical Methods

ICP-emission spectrometry (ICP-OES) and ICP-mass spectrometry were used to analyze samples for main oxides, trace, and rare earth elements at Acme Analytical Laboratories in Vancouver, Canada (ICP-MS). Major oxides and trace elements have detection limits of 0.001–0.04 wt.% and 0.01–0.5 ppm, respectively. Analytical precision is 0.5% for major elements and 2 to 20% for trace elements, according to duplicate analyses.

## 3. Results and Discussion

### 3.1. General Geochemical Characteristics of the Studied Samples

Table 1 shows the chemical compositions of the granitic rocks in the research area. The altered granites analyzed had high silica, titanium, and phosphorous content, as well as increased levels of trace elements, such as Nb, Ta, Zn, Mo, Pb, and Re, and increased levels of REE, particularly light ones.

### 3.2. Geochemistry of Major and Trace Elements

According to Nockolds et al., 1979 and El Kholy et al. (2017), residual fluids rich in water modify granites to different degrees, and these rocks then undergo deuteric and hydrothermal modifications [23,24]. The majority of granitic rocks are susceptible to alteration, particularly after emplacement and solidification. The distribution of trace elements is determined by the main oxides; i.e., it is also influenced by the mineral composition and rock formation conditions. During the melting and crystallization of magma, several incompatible trace elements (e.g., Ba, Rb, Sr, Nb, Ta, Zr, U, Th, and Y) tend to concentrate in the liquid and late-stage phases, and these elements are compatible elements in the peraluminous granites.

As a result, they are typically enriched in granitic liquids [25]. These elements, like other forms of trace elements, are mainly encountered in the late phases of magmatic differentiation as suspended and/or distributed impurities (Table 1). The loss and gain of elements causes geochemical changes in major and trace elements. According to Rudnick and Gao (2003), normalizing altered granites on the upper continental crust (UCC) is advised for understanding the geochemical behavior of major, trace elements in the studied granites [26].

Afterward, the reference UCC becomes flat at unity, and the relative depletion or enrichment is given by the deviations on both sides of the reference line (Figure 3). Geochemistry of major elements is discussed in terms of gains (positive) and losses (negative) of these elements during granite alteration. Most altered granite samples exhibit an increase in $SiO_2$, $TiO_2$, and $P_2O_5$, in addition to a noticeable decrease in $Al_2O_3$, $Fe_2O_3$, MgO, CaO, $Na_2O$, $K_2O$, and MnO in all samples.

Enrichment in $SiO_2$ may be correlated with the silicification processes, as indicated from petrographic studies [27], and can be due to the considerable quantity of free Si-bearing mineral phases (e.g., quartz) that increase in the more evolved granites. The silicification process leads to silica enrichment at the expense of other major oxides. The formation of titanite, rutile, and ilmenite is reflected by increased $TiO_2$ contents in all of the altered granite samples' alteration of silicate minerals (feldspars and ferromagnesians) and formation of Ti-bearing minerals. Enhanced values of $P_2O_5$ could be connected to the presence of apatite, monazite, and xenotime [28].

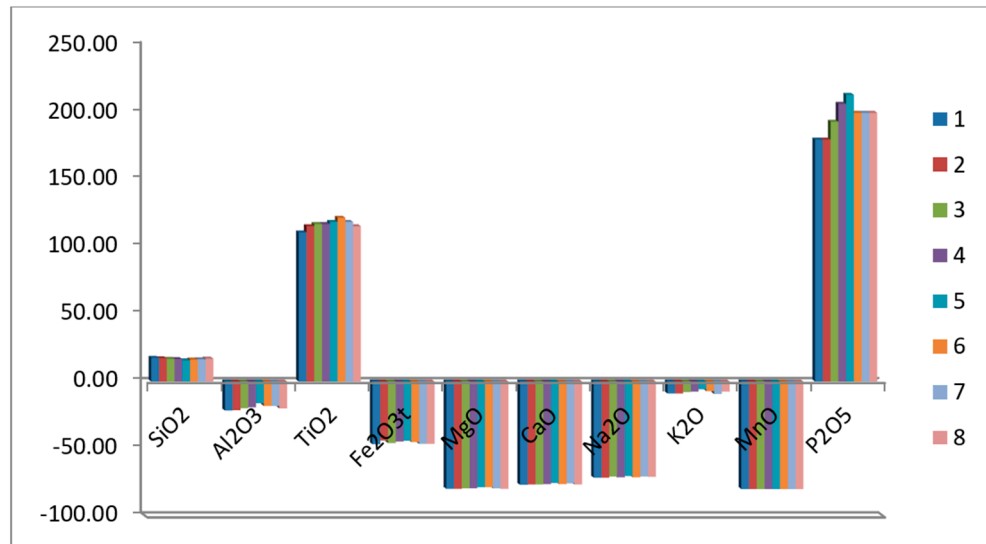

**Figure 3.** Depletion-enrichment variation diagram of major oxides in the studied altered granites.

The depletion of Na, K, and Ca in all samples (Figure 3) could be related to feldspar alteration and the release of these elements into the fluids responsible for alteration. The loss of Mg and Mn in all of the studied samples may result from the destruction and alteration of ferromagnesian minerals.

The presence of enrichment-depletion trends for minor elements (Figure 4) is due to the variations of the physico–chemical conditions of the fluids, such as the pH, Eh, temperature, and ionic complexes during the development of this deposit [29]. Abundances of trace elements in the hydrothermally altered granite samples are either higher (Ga, Hf, Nb, Rb, Sn, Sr, Ta, Th, W, Zr, Mo, Pb, Zn, Bi, As, Cd, Sb, Be, Li, In, Re, Y, U, and Ag) or lower (Ba, Co, Cs, V, Cu, Ni, Cr, and Sc) than those in the upper continental crust.

Depletion in Ba and Cs could be related to feldspar decomposition. The loss of Co, Cr, Ni, and Cu could be connected to the destruction of the ferromagnesian minerals by altering the solutions [30]. Rb, Mo, and Ga increase with increasing sericite clay minerals due to the sericitization processes, which may be related to the loss of Ba. It was suggested that Rb concentrations increased in a liquid-rich phase [31].

**Table 1.** Major oxides, trace, and rare earth elements with some of their ratios and the calculated tetrad effect of the studied altered granites.

| Oxides | El Sella Granites | | | | | | | |
|---|---|---|---|---|---|---|---|---|
| % | 1 | 2 | 3 | 4 | 5 | 6 | 7 | 8 |
| $SiO_2$ | 78.36 | 78.13 | 77.88 | 77.73 | 77.06 | 77.55 | 77.66 | 77.97 |
| $Al_2O_3$ | 12.07 | 12.11 | 12.34 | 12.45 | 12.91 | 12.60 | 12.60 | 12.32 |
| $TiO_2$ | 1.35 | 1.38 | 1.39 | 1.39 | 1.4 | 1.42 | 1.4 | 1.38 |
| $Fe_2O_3{}^t$ | 2.69 | 2.83 | 2.73 | 2.79 | 2.82 | 2.77 | 2.70 | 2.7 |
| MgO | 0.5 | 0.5 | 0.51 | 0.51 | 0.53 | 0.53 | 0.51 | 0.5 |
| CaO | 0.83 | 0.84 | 0.84 | 0.85 | 0.88 | 0.85 | 0.87 | 0.84 |
| $Na_2O$ | 0.93 | 0.93 | 0.96 | 0.94 | 0.97 | 0.94 | 0.96 | 0.95 |
| $K_2O$ | 2.55 | 2.55 | 2.58 | 2.59 | 2.64 | 2.60 | 2.54 | 2.59 |
| MnO | 0.02 | 0.02 | 0.02 | 0.02 | 0.02 | 0.02 | 0.02 | 0.02 |
| $P_2O_5$ | 0.42 | 0.42 | 0.44 | 0.46 | 0.47 | 0.45 | 0.45 | 0.45 |
| $SO_3$ | 0.15 | 0.17 | 0.17 | 0.17 | 0.17 | 0.17 | 0.17 | 0.15 |
| L.O.I. | 0.13 | 0.12 | 0.14 | 0.1 | 0.13 | 0.1 | 0.12 | 0.127 |
| Total | 100 | 100 | 100 | 100 | 100 | 100 | 100 | 100 |



**Table 1.** *Cont.*

| Oxides | El Sella Granites | | | | | | | |
|---|---|---|---|---|---|---|---|---|
| % | 1 | 2 | 3 | 4 | 5 | 6 | 7 | 8 |
| Ba | 82 | 85 | 86 | 90 | 88 | 86 | 83 | 85 |
| Co | 3.2 | 3.4 | 3.2 | 3.4 | 3.4 | 3.2 | 3.1 | 3.2 |
| Cs | 4 | 4.1 | 4 | 4 | 4.3 | 4.2 | 4.1 | 4 |
| Ga | 26.68 | 28.01 | 27.14 | 26.7 | 27.56 | 26.61 | 26.83 | 26.9 |
| Hf | 7.93 | 8 | 8.24 | 7.76 | 7.44 | 7.53 | 7.53 | 7.77 |
| Nb | 90.82 | 90.64 | 90.64 | 87.21 | 88.38 | 87.02 | 89.07 | 86.56 |
| Rb | 202.9 | 202.7 | 204.9 | 205.7 | 212.3 | 206.2 | 209.1 | 207.3 |
| Sn | 3.4 | 3.5 | 3.6 | 3.2 | 3.3 | 3.2 | 3.3 | 3.3 |
| Sr | 576 | 589 | 597 | 601 | 613 | 587 | 597 | 584 |
| Ta | 4.4 | 4.3 | 4.1 | 4.2 | 4.1 | 4 | 4.2 | 3.9 |
| Th | 23.2 | 21.7 | 21.1 | 21.2 | 20.9 | 20.5 | 21 | 21.1 |
| Tl | 1.58 | 1.61 | 1.64 | 1.63 | 1.7 | 1.62 | 1.63 | 1.66 |
| U | 2049.7 | 2093.1 | 2091.7 | 2044.9 | 2161.5 | 2131.3 | 2093.8 | 2105 |
| V | 58 | 59 | 59 | 59 | 61 | 60 | 60 | 60 |
| W | 5.3 | 4.8 | 5.2 | 5.1 | 5.5 | 5 | 4.8 | 4.8 |
| Zr | 268.9 | 278.7 | 273.6 | 267 | 270.7 | 273.5 | 270.9 | 266.4 |
| Mo | 4.16 | 5.08 | 4.43 | 4.81 | 4.63 | 4.22 | 4 | 4.4 |
| Ag (ppb) | 3541 | 3354 | 3274 | 3558 | 3687 | 3543 | 3503 | 3466 |
| Cu | 12.2 | 11.8 | 11.2 | 9.6 | 12.8 | 9.7 | 9.8 | 12.2 |
| Pb | 156.31 | 164.77 | 163.02 | 162.03 | 165.82 | 163.07 | 162.52 | 165.05 |
| Zn | 193.2 | 198.6 | 196.8 | 188.9 | 208.5 | 197.8 | 194.4 | 192.6 |
| Ni | 7.7 | 8.3 | 7.2 | 7.4 | 8 | 8.5 | 6.8 | 7.3 |
| Bi | 0.29 | 0.33 | 0.3 | 0.32 | 0.31 | 0.31 | 0.3 | 0.3 |
| As | 10.6 | 11.5 | 11 | 11 | 11.1 | 11.2 | 10.9 | 11.2 |
| Cd | 0.17 | 0.23 | 0.17 | 0.15 | 0.17 | 0.08 | 0.18 | 0.18 |
| Sb | 1.01 | 1.08 | 1.09 | 1.05 | 1.08 | 1.08 | 1.12 | 1.04 |
| Cr | 4 | 4 | 4 | 5 | 4 | 4 | 4 | 4 |
| Be | 6 | 7 | 7 | 6 | 7 | 7 | 6 | 7 |
| Sc | 8 | 8.4 | 8.7 | 8.1 | 8.7 | 8.4 | 8.6 | 8.2 |
| Li | 28 | 27.2 | 26.8 | 26.8 | 27.8 | 27.6 | 27.9 | 25.8 |
| In | 0.06 | 0.07 | 0.07 | 0.07 | 0.06 | 0.07 | 0.05 | 0.07 |
| Re | <0.002 | 0.002 | <0.002 | <0.002 | <0.002 | <0.002 | <0.002 | <0.002 |
| Y | 47.1 | 49.1 | 47 | 46.6 | 48.9 | 47.9 | 47.6 | 45.9 |
| La | 112.6 | 115.8 | 117.7 | 115.5 | 120.7 | 118.6 | 117.7 | 115.6 |
| Ce | 166.61 | 172.11 | 171.57 | 171.85 | 178.72 | 170.46 | 170.46 | 170.46 |
| Pr | 29.8 | 31.4 | 30.9 | 30.7 | 32.2 | 30.8 | 31 | 30.5 |
| Nd | 103 | 104.8 | 101.4 | 100 | 106.9 | 103.3 | 104.3 | 102 |
| Sm | 17.6 | 17.5 | 17.1 | 17.4 | 17.7 | 17.7 | 17.3 | 17.1 |
| Eu | 2 | 1.8 | 1.9 | 2 | 1.9 | 1.8 | 2 | 1.9 |
| Gd | 11.4 | 12.4 | 12.5 | 12.2 | 12 | 12 | 11.9 | 11.2 |
| Tb | 1.5 | 1.5 | 1.5 | 1.5 | 1.5 | 1.5 | 1.4 | 1.5 |
| Dy | 7.4 | 7.1 | 7.6 | 7.7 | 7.3 | 7.1 | 7.3 | 7.3 |
| Ho | 1.4 | 1.4 | 1.3 | 1.4 | 1.4 | 1.2 | 1.2 | 1.2 |
| Er | 3.7 | 3.7 | 3.8 | 3.7 | 3.8 | 3.8 | 3.6 | 3.6 |
| Tm | 0.6 | 0.6 | 0.5 | 0.5 | 0.6 | 0.5 | 0.5 | 0.5 |
| Yb | 3.8 | 3.6 | 3.6 | 3.5 | 3.5 | 3.5 | 3.3 | 3.3 |
| Lu | 0.5 | 0.5 | 0.5 | 0.5 | 0.5 | 0.5 | 0.5 | 0.5 |
| REE | 461.91 | 474.21 | 471.87 | 468.45 | 488.72 | 472.76 | 472.46 | 461.91 |
| La/Yb$_n$ | 29.63 | 32.17 | 32.69 | 33.00 | 34.49 | 33.89 | 35.67 | 35.03 |
| La/Sm$_n$ | 6.40 | 6.62 | 6.88 | 6.64 | 6.82 | 6.70 | 6.80 | 6.76 |
| La/Y | 2.39 | 2.36 | 2.50 | 2.48 | 2.47 | 2.48 | 2.47 | 2.52 |
| La/Ta | 25.59 | 26.93 | 28.71 | 27.50 | 29.44 | 29.65 | 28.02 | 29.64 |
| Gd/Yb$_n$ | 3 | 3.44 | 3.47 | 3.49 | 3.43 | 3.43 | 3.61 | 3.39 |
| Sr/Eu | 288 | 327.22 | 314.21 | 300.50 | 322.63 | 326.11 | 298.50 | 307.37 |
| Eu/Sm | 0.11 | 0.10 | 0.11 | 0.11 | 0.11 | 0.10 | 0.12 | 0.11 |
| Y/Ho | 33.64 | 35.07 | 36.15 | 33.29 | 34.93 | 39.92 | 39.67 | 38.25 |

**Table 1.** *Cont.*

| Oxides | El Sella Granites | | | | | | | |
|---|---|---|---|---|---|---|---|---|
| % | 1 | 2 | 3 | 4 | 5 | 6 | 7 | 8 |
| LREE | 431.61 | 443.41 | 440.57 | 437.45 | 458.12 | 442.66 | 442.76 | 437.56 |
| HREE | 30.3 | 30.8 | 31.3 | 31 | 30.6 | 30.1 | 29.7 | 29.1 |
| LREE/HREE | 14.24 | 14.40 | 14.08 | 14.11 | 14.97 | 14.71 | 14.91 | 15.04 |
| Ce/Ce* | 0.68 | 0.69 | 0.68 | 0.7 | 0.69 | 0.67 | 0.67 | 0.68 |
| Eu/Eu** | 0.7 | 0.6 | 0.6 | 0.6 | 0.6 | 0.6 | 0.7 | 0.6 |
| Zr/Hf | 33.91 | 34.84 | 33.20 | 34.41 | 36.38 | 36.32 | 35.98 | 34.29 |
| Nb/Ta | 20.64 | 21.08 | 22.11 | 20.76 | 21.56 | 21.76 | 21.21 | 22.19 |
| U/Th | 88.35 | 96.46 | 99.13 | 96.46 | 103.42 | 103.97 | 99.70 | 99.76 |
| Ba/Sr | 0.14 | 0.14 | 0.14 | 0.15 | 0.14 | 0.15 | 0.14 | 0.15 |
| Ba/Rb | 0.40 | 0.42 | 0.42 | 0.44 | 0.41 | 0.42 | 0.40 | 0.41 |
| Rb/Sr | 0.35 | 0.34 | 0.34 | 0.34 | 0.35 | 0.35 | 0.35 | 0.35 |
| $t_3$ | 0.97 | 0.91 | 0.98 | 0.96 | 0.94 | 1 | 0.99 | 1.05 |
| $t_4$ | 1.12 | 1.09 | 0.99 | 0.98 | 1.07 | 0.97 | 0.96 | 0.96 |

Ce*: Ce anomaly. Eu**: Eu anomaly.

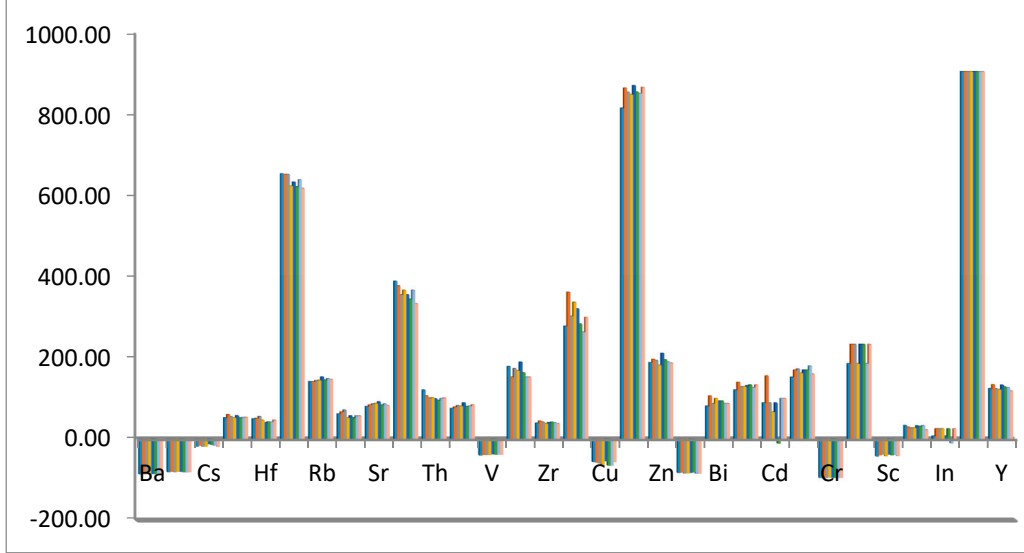

**Figure 4.** Depletion-enrichment variation diagram of trace elements in the studied altered granites.

While zirconium enrichment was relative to the continental crust, Zr showed reduction in its contents compared with that of the other granitic rocks. This diminution could be linked to zircon fractionation since it is an early crystallized mineral. High U, Nb, Y, and Pb contents may be controlled by the presence of uranophane, uranothorite, autunite, kasolite, columbite, xenotime, and zircon. Sr increasing could be linked to the calcium in fluorite, resulting from fluoritization processes. Silver enrichment is related to the presence of pyrite-bearing silver [32].

Rhenium is one of the rarest and most dispersed metallic elements in the earth's crust, with abundance estimated to be about 1 part per billion. Although traces of rhenium occur in some minerals, molybdenite is the only significant host mineral. Rhenium enrichment in the altered granites we studied may be connected with the abundance of Mo, while an increase in indium could be linked to pyrite. Increments in gold and silver could be related to their presence as separate grains or associated with sulphides. El Mezayen et al. (2019) reported the presence of gold, with concentrations reaching up to 1 ppm [7].

## 4. Alteration Processes in the Studied Altered Granites

Sul Hamed altered granites are characterized by silicification and desilicification processes, which are usually accompanied by an increase or decrease in the other elements. This enrichment could merely be a consequence of quartz leaching. El Feky et al. (2011) stated that granitic rocks are generally altered after solidification and emplacement due to the effect of residual solutions, which are rich in water and cause deuteric and hydrothermal alteration processes [33].

According to the $K_2O/Na_2O$ versus $SiO_2/Al_2O_3$ diagram provided by Recio et al. (1997) [34], most samples lie along the trend of a K-feldspar addition due to K-metasomatism or around a silica-addition trend (Figure 5a). In the Q–P diagram (Figure 5b) provided by Cunny et al. (1989) [35], we observed two distinct reaction processes, either magmatic differentiation or hydrothermal alterations: The first hydrothermal alteration was dequartzification–albitization, toward an albitization increase of albite with silica depletion. Another alteration trend appeared toward K-metasomatism and an increase in quartz and secondary silica formation. Most samples of the altered granite occurred around K-metasomatism trend, while one sample had a Na-metasomatism trend (Figure 5c). Cunney et al. (1989) [35] constructed a Na-K diagram that can identify different alteration zones, such as argillic, potassic, Sodic, and dequartzification alterations (silica dissolution) (Figure 5c). The K-metasomatism field is restricted to the left segment of the diagram, where the K increases and Na decreases. The right segment of the diagram shows the Na-metasomatism field, where the Na increases and K decreases. The argillic alteration field was restricted to the lower part of the Na-K binary diagram that indicated the high Na- and K-leaching conditions of altered rocks. Most samples show that the altered granites we studied are affected by K-metasomatism with argillization. On the Q-F1 diagram of Cathelineau (1986) [36], some samples indicate the criteria of a silica addition with fluoritization, in contrast to the other silica addition, which suffered from illitization (Figure 5d).

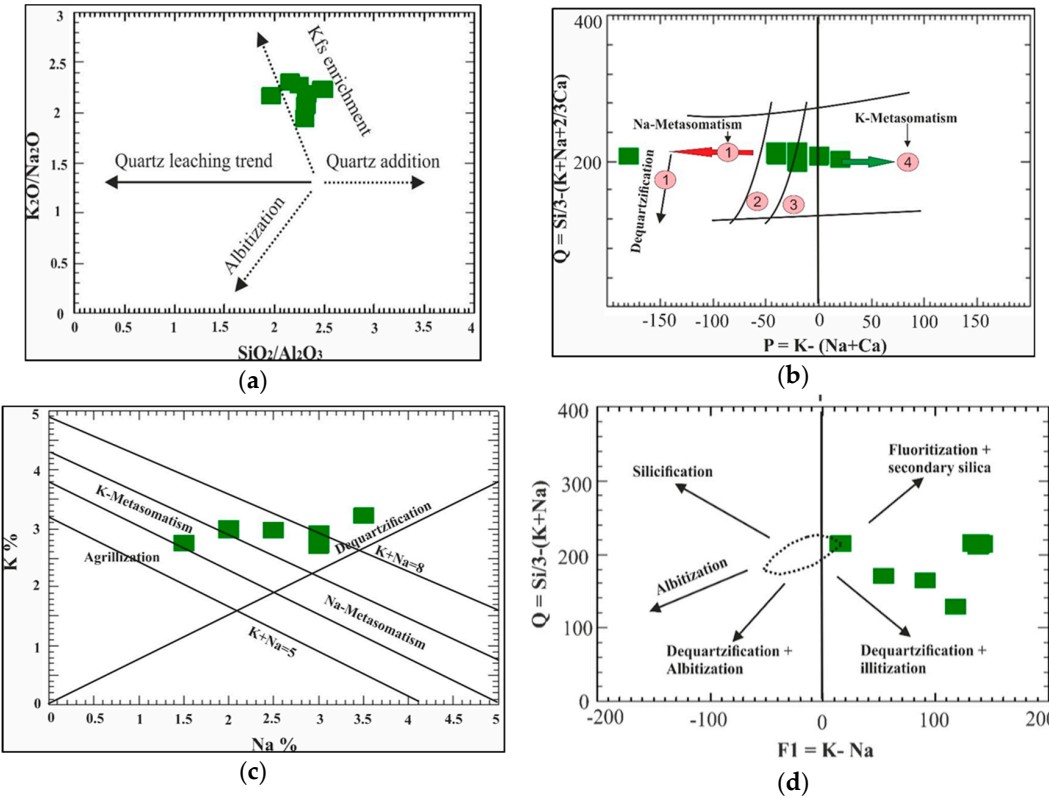

**Figure 5.** (**a**) $K_2O/Na_2O$ versus $SiO_2/Al_2O_3$ binary diagram plots for the studied altered granite [34]. (**b**) Q-P diagram of Cunny et al. (1989) [35]. (**c**) Na-K diagram of Cunny et al. (1989) [35]. (**d**) Q1 vs. F1 binary diagram plots for the studied altered granite [36].

## 5. Geochemistry of Isovalents and Related Elements

Typically, the tetrad effect is accompanied by a modification in trace elements' behavior [37,38]. The modified geochemical behavior of many trace elements is known as non-CHARAC behavior, which mainly includes highly evolved magmatic systems rich in $H_2O$, $CO_2$, and elements such as Li, B, F, and/or Cl, and which may be regarded as transitional between a pure silicate melt and an aqueous fluid [37]. Zr and Hf have very similar geochemical behavior, which results in the determined ratios in some geological materials (Zr/Hf ratios from 33–40) found in most igneous rocks. Deviation from this ratio is rare and revised to metasomatism or intense fractionation of accessory minerals [14].

The Zr/Hf ratio of the studied altered granites lies between 33.2 and 36.38, with 34.89 as an average, which is in the range of chondritic ratios. The Y/Ho ratio deviates from the chondritic value (Y/Ho = 28.8) only in samples showing a tetrad effect [39]. The complexation with bicarbonate is assumed to generate Y/Ho values < 28, while the complexation with fluorine is interpreted as a major cause for Y/Ho > 28.

The studied samples have Y/Ho ratios higher than the chondritic value (33.29–39.95), suggesting the complexation with fluorine, which was confirmed by the fluorite presence [40]. The studied altered granites also show non-chondritic ratios for Nb/Ta (20.64–22.19), La/Nb (1.24–1.37), and La/Ta (25.59–29.64), indicating the highly differentiated nature of the studied altered granites. The chondritic ratios are 17.6 ± 1 for Nb/Ta, (0.96–1) for La/Nb, and (16–18) for La/Ta [41,42].

The Rb/Sr ratio increases with differentiation; this is due to the depletion of Sr in the liquid magma as a result of crystallization of feldspar, while Rb is enriched in the liquid phase (in highly differentiated granite, the Rb/Sr ratio is usually high). The studied altered granites have low Rb/Sr ratios (0.34–0.35), suggesting Sr enrichment during fluoritization. The very high U/Th ratio (88.35–103.97), which was much larger than the chondritic ratio (0.33), could be another indication of the highly evolved nature of these rocks.

## 6. Rare Earth Elements

The average of the total REE content of the studied altered granite (av. ΣREE = 472.77 ppm) is higher than that of the worldwide granite (ΣREE = 250–270 ppm) as given by Hermann (1970) [41]. The enrichment of REE could be attributed to hydrothermal alteration. In the studied granite, there is an increase in the REE in general, where ∑REE 461.91–488.72 ppm (Table 1), but ∑LREE 431.61–458.12 ppm is higher than ∑HREE 29.1–31.1 ppm and ∑LREE/∑HREE 14.08–15.04. The fractionation of the REE in general is high for La/Yb (29.63–35.67). La/Yb is well-known to range between (30 and 80) for metaluminous or slightly peraluminous granites (Figure 6).

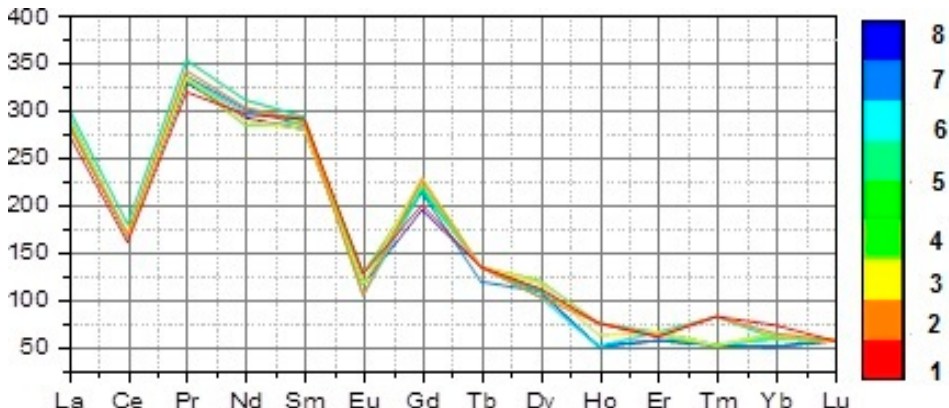

**Figure 6.** UCC-normalized REE diagram [26] for the studied granites.

The LREE (La/Sm 6.4–6.9) and the HREE (Gd/Yb 3.00 to 3.61) are moderately fractionated. A negative Eu anomaly characterizes this granite, where Eu/Eu* (0.6–0.7). The

fractionation among Eu and neighboring REE (Sm and Gd) with the tetrad effect possibly leads to a decrease in the magnitude of the negative Eu anomaly [43,44].

The studied altered granites have a peculiar pattern of the rare earth chondrite-normalized elements (Figure 4). An M-type of REE tetrad effect is in the $t_3$ and $t_4$ types [5,45].

The first and the third tetrad are not recognized due to the presence of a negative Ce anomaly in t1 and the low calculated values (<1.1) in t3. The second tetrad is comparably difficult to recognize due to the anomalous behavior of Eu and the fact that Pm does not occur in nature. Figure 5 shows that only one sample has a strong M-type in the fourth tetrad.

Variation diagrams of U and some trace elements (Figure 7) illustrated that U has a weak positive correlation with Y (Figure 7d) and a strong positive correlation with gold (Figure 7h), while it has a weak to moderate negative correlation with Hf and Zr/U (Figure 5d,f). In addition, uranium has a weakly defined correlation with the other trace elements, indicating a weak to moderate effect of magmatic processes, while the post-magmatic processes or underground water could greatly influence the redistribution of uranium and other elements.

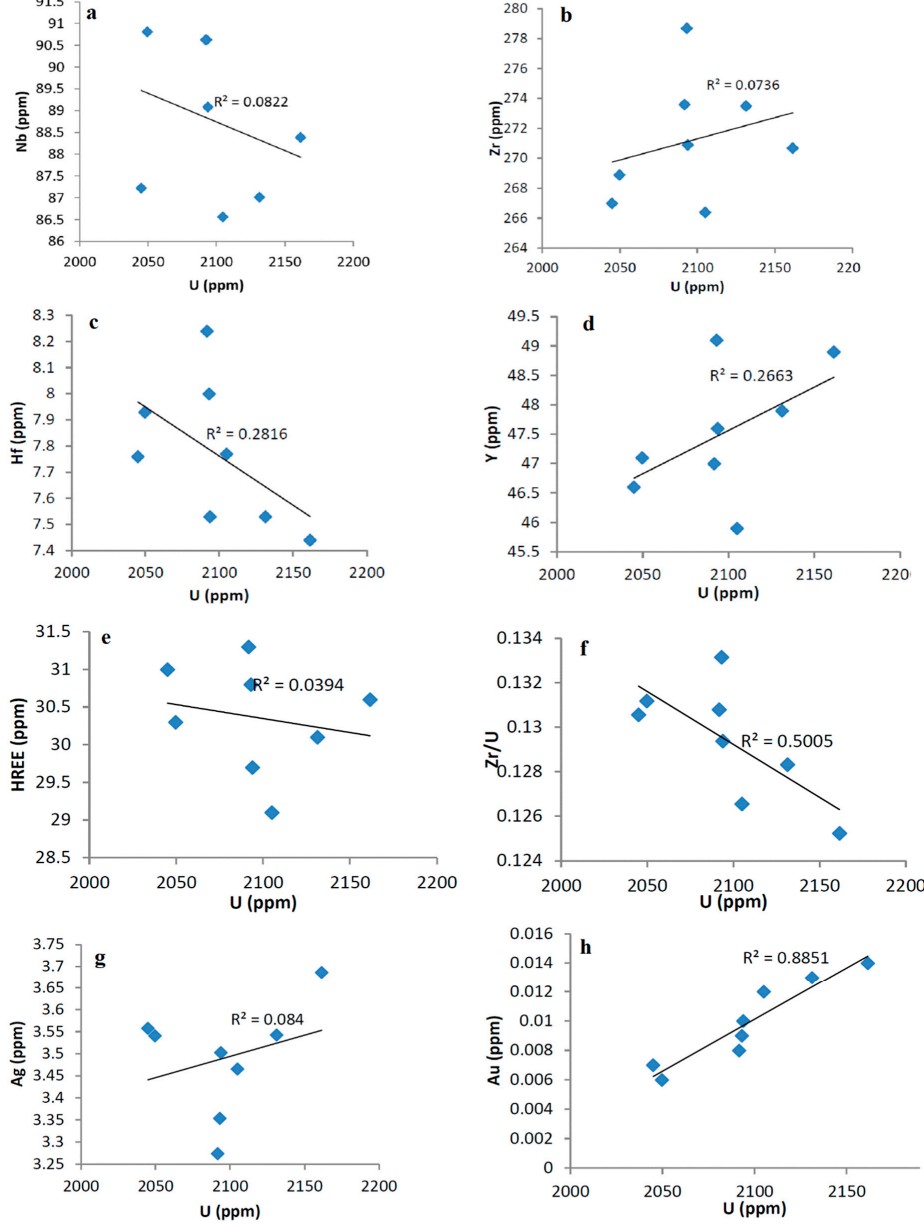

**Figure 7.** Variation diagrams of U and some trace elements (**a**–**h**). Blue squares mean altered granite samples.

## 7. Conclusions

Kaolinization, silicification, and hematitization are the main alteration processes and are associated with uranium and other mineralization. The studied altered granites have high silica, titanium, and phosphorous, with enhanced amounts of trace elements such as Nb, Ta, Zn, Mo, Pb, and Re in addition to REE, especially light ones. An enrichment in $SiO_2$ may be correlated with the silicification processes and the considerable quantity of free Si-bearing mineral phases that increase in the more evolved granites. The formation of titanite, rutile, and ilmenite is reflected by increased $TiO_2$ contents in all altered granite samples and the alteration of feldspars and ferromagnesian minerals and the formation of Ti-bearing minerals. Enhanced values of $P_2O_5$ could be connected with the presence of apatite, monazite, and xenotime. The depletion of Na, K, and Ca in all samples could be related to feldspar alteration and the release of these elements into the fluids responsible for alteration, while the loss of Mg and Mn may result from the destruction and alteration of ferromagnesian minerals. High U, Nb, Y, and Pb contents may be controlled by the presence of uranophane, uranothorite, autunite, kasolite, columbite, xenotime, and zircon. A Sr increasing could be linked to fluoritization processes. The increments in gold and silver could be related to their presence as separate grains or associated with sulphides. Rb, Mo, and Ga increased with increasing sericite and clay minerals due to sericitization processes. Additionally, Rb concentrations increased in a liquid-rich phase. Depletion in Ba, Cs, Co, Cr, Ni, and Cu could be related to the feldspar and ferromagnesian minerals' decomposition. Rarest elements increasing were recorded for the first time in these rocks. The rhenium enhancement in the studied altered granites may be connected with the abundance of Mo in sulphides relative to rhenium, while the increase in indium could be linked to pyrite. The average of the total REE content of the studied altered granite was higher than that of the worldwide granites with LREE enrichment. A strong M-type tetrad and weak W-type indicated the effect of hydrothermal alteration processes in the altered granites, and they were confirmed by the calculated ratios of most isovalents that deviated from the chondritic ratio in many values. Variation diagrams of U and some trace elements illustrated a weak to moderate effect of magmatic processes, while the post-magmatic processes or underground water could greatly influence the redistribution of uranium and other elements mineralization.

**Author Contributions:** Conceptualization, N.S.A., M.G.E.F., M.R.K. and M.Y.H.; methodology, N.S.A., M.G.E.F. and M.R.K.; software, E.E.S.M. and M.Y.H.; validation, A.E.-T., M.G.E.F., M.S.A. and E.S.Y.; formal analysis, M.G.E.F. and M.Y.H.; investigation, M.G.E.F., A.E.-T. and E.E.S.M.; resources, N.S.A., M.G.E.F. and M.R.K.; data curation, M.G.E.F. and M.Y.H.; writing—original draft preparation, N.S.A., M.G.E.F. and M.R.K.; writing—review and editing, N.S.A., M.G.E.F., M.R.K. and M.Y.H.; visualization, N.S.A., A.E.-T. and E.E.S.M.; supervision, M.G.E.F., A.E.-T., M.S.A. and E.S.Y.; project administration, N.S.A., M.G.E.F. and M.R.K.; funding acquisition, M.S.A. and E.S.Y. All authors have read and agreed to the published version of the manuscript.

**Funding:** The authors extend their appreciation to the Deanship of Scientific Research at King Khalid University (KKU) for funding this research project Number (R.G.P.2/247/43).

**Data Availability Statement:** Not applicable.

**Acknowledgments:** The authors extend their appreciation to the Deanship of Scientific Research at King Khalid University (KKU) for funding this research project Number (R.G.P.2/247/43).

**Conflicts of Interest:** The authors declare no conflict of interest.

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
