# Peer review of "Geochemical Conditions and Factors Controlling the Distribution of Major, Trace, and Rare Elements in Sul Hamed Granitic Rocks, Southeastern Desert, Egypt"

_minerals, doi:10.3390/min12101245_

Round 1

Reviewer 1 Report

The presented chemical composition of altered granites is really very interesting. What is especially missing: any comparison with unaltered granites of the same massif– if it cannot be directly presented, at least some verbal information with references to literature should be added. Also any information about mineralogy and petrography is missing – therefore, it is unclear to what extent authors really observed the minerals mentioned, and whether some minerals (e.g., fluorite) are only supposed due to rock‘s chemistry.

I disagree with authors about the tetrad effect: there is no W-type in the first tetrad – instead, there is negative cerium anomaly (thus the value t1 cannot be calculated, but La, Pr and Nd seem to have form the M-type first tetrad). So, there is only the first tetrad (probably) and second tetrad (possibly), both of M type.

Regarding the combination of high SiO2 and TiO2: I suggest to consider (discuss) also the possibility that TiO2 was not enriched by alteration but instead, the rocks could have originally rather granodioritic composition and then they were silicified (with significant removal of Ca, Mg, Na and partly Fe, but not Ti; the same as for Ti holds for Zr, Hf and Sc); the little mobile elements (Ti,Zr,Hf…) could be also somewhat passively concentrated.

Also information about ages of the granites and of the uranium mineralization would be useful (it would also show whether the high Pb content reflects hydrothermal enrichment, or majority of the lead formed by uranium decay).

Unfortunately the manuscript suffers from little consistent presentation and interpretation of data. „Enhanced“ rhenium is mentioned several times in the text (and high Re is also displayed in Fig. 3), but in Table 1, all samples have Re below detection limit. It is unclear what starting composition of granites was used for the charts in Fig. 4; but if there was net desilification / dequartzification, most of the original granites would have at least 80 % SiO2 which is impossible.

I did not suggest language corrections where I understand, but an improvement is necessary.

Particular comments:

row 62: Tyler instead of „Tylor“

row 70-71: „REE … have higher stability“: correctly perhpas „REE compounds (complexes)…“

row 72-74: note that important REE-bearing rockforming minerals, monazite and xenotime, may also display W-type tetrad effect – rather weak but sufficient to cause M-type tetrad effect in the remaining melt (e.g., Tin and Keppler, 2015; Stepanov et al., 2012).

row 76: „according to one study“ – what study?

row 81: „Y with no 4f electrons is not a pseudo-lanthanide…“ – I think more logical would be: „Y with no 4f electrons IS a pseudo-lanthanide…“ (or behaves like a (pseudo)lanthanide…)

row 88: „rarest elements“ – please specify (I think from the elements mentioned Au, Re and In)

row 126-127: note that Sr, Ba and Zr, usually also Th are rather compatible in peraluminous granites

rows 140-145: note that the granite composition prior to hydrothermal alteration did not have to be identical as the mean upper continental crust. The Figures 2 and 3 are nice but I think that formulations could be modified.

rows 150-152: is there any evidence (e.g., from relations observed with petrographic microscope or SEM) that the Ti-rich minerals are products of alteration? Note that if Ti-oxides are formed by alteration of biotite (from Ti contained in biotite) it does not enhance Ti content of the rock (or enhances only slightly by passive concentration, if MgO and other compounds are partly removed).

row 162: regarding Pb, the information about the age of uranium mineralization is important. If it is older than ca. 200-250 millions years, most of the lead would have formed by uranium decay.

row 164: micas are usually more important pool of Cs than feldspars (and during alteration, they liberate Cs easily)

(Table 1): very low LOI! (is it from ignition at 1000 or 1100 °C, as usual in igneous rocks, or at lower temperature?) Totals of major component are not shown (or are the values normalized to the sum 100 %?)

rows 182,306,307: „rhenium enhancement…“ – what enhancement? In all analyses shown, Re is below detection limit. For the same reason I have doubts if Re in Figure 3 is correct.

row 188: what is the evidence for desilicification? It is difficult to imagine that the rocks contained more than 78 % SiO2 (the actual content) prior to the alteration.

rows 190, 192: the cited reference (El Feky et al.) is missing. „residual solutions which are rich in water“ – perhaps better „residual aqueous solutions“?

rows 198, 201: „dequatzification“ („r“ missing)

Fig. 4: I do not know what units are used, but in the first chart, the ratio SiO2/Al2O3 is between ca. 2 and 2.5. This contradicts the chemical composition of rocks (from values in Table 1: mass ratio SiO2/Al2O3 is ca. 6, and molar ratio ca. 10, or Si/Al molar ratio ca. 5). Please check the units and calculations in other charts, too.

row 216: note that the pairs La-Nb, La-Ta and Rb-Sr do not represent isovalent elements. The title should be modified.

row 224, 225: I would not say that deviation of Zr/Hf ratios from the range 33-40 are rare: lower values are common in fractionated magmas. In any case, geochemical or petrological, not technical literature should be cited here.

rows 227,241: „natural“ ratio – replace by „chondritic ratio“ (samples with non-chondritic ratios are natural, too)

rows 239, 240, 301: is there any evidence of fluorite presence?

Chapter 6. Rare Earth Elements“, page 10 (general comment): the major irregularity of the REE distribution is negative cerium anomaly. It is probably combined with a weak M-type tetrad effect forming the first tetrad (and probably the second one – which would be better seen in a chart with the space for promethium, however it is uncertain due to Eu anomaly). Any significant third or fourth tetrad is not observable; in addition, the values t3 and t4 in Table 1 are usually between 0.90 and 1.10, which means that tetrad effect is not significant (t1 cannot be used due to obvious Ce anomaly). The negative Ce anomaly indicates that important portion of trivalent REE (mainly LREE) was brought in an oxidizing solution (depleted in tetravalent cerium which is less soluble), probably together with uranium.

row 266: „Figure 19“??

Figure 6: the chart Ta-U is repeated. Please check the R2 values: I do not believe that R2 = 0,90 in the chart U – Zr/U is correct!

rows 281,282: the correlation of U and Hf in the chart is weakly negative. Nevertheless, most correlations are insignificant due to low chemical variability of the samples presented. Comparison with little altered or unaltered local granites would be much more interesting.

row 300: 2x „kasolite“; were the uranium minerals documented?

row 304: „liquids rich phase“??

rows 308-309: the sentence „Increments of gold…“ is repeated (see r. 301-302)

Author Response

Dear Reviewer,

Please find attached the submission of the carefully revised version of the manuscript in Ref., following the minor comments and modifications of the Reviewers.

Below, the detailed list of the changes made in response to the Reviewer’s minor comments (in italics), outlining every change made a point by point, is provided. The changes are marked in the manuscript text in the colour highlighted for the reviewer.

Comments and Suggestions for Authors

The presented chemical composition of altered granites is really very interesting. What is especially missing: any comparison with unaltered granites of the same massif– if it cannot be directly presented, at least some verbal information with references to literature should be added. Also any information about mineralogy and petrography is missing – therefore, it is unclear to what extent authors really observed the minerals mentioned, and whether some minerals (e.g., fluorite) are only supposed due to rock‘s chemistry.

I disagree with authors about the tetrad effect: there is no W-type in the first tetrad – instead, there is negative cerium anomaly (thus the value t1 cannot be calculated, but La, Pr and Nd seem to have form the M-type first tetrad). So, there is only the first tetrad (probably) and second tetrad (possibly), both of M type.

Regarding the combination of high SiO2 and TiO2: I suggest to consider (discuss) also the possibility that TiO2 was not enriched by alteration but instead, the rocks could have originally rather granodioritic composition and then they were silicified (with significant removal of Ca, Mg, Na and partly Fe, but not Ti; the same as for Ti holds for Zr, Hf and Sc); the little mobile elements (Ti,Zr,Hf…) could be also somewhat passively concentrated.

Also information about ages of the granites and of the uranium mineralization would be useful (it would also show whether the high Pb content reflects hydrothermal enrichment, or majority of the lead formed by uranium decay). coronadite as Mn-Pb mineral.

Unfortunately the manuscript suffers from little consistent presentation and interpretation of data. „Enhanced“ rhenium is mentioned several times in the text (and high Re is also displayed in Fig. 3), but in Table 1, all samples have Re below detection limit.

Re concentration in ppm in our study, where its natural contets don,t exceed ppb fractions in the same rocks.  It is unclear what starting composition of granites was used for the charts in Fig. 4; but if there was net desilification / dequartzification, most of the original granites would have at least 80 % SiO2 which is impossible.

I did not suggest language corrections where I understand, but an improvement is necessary.

Response: the authors accept with your comments and suggestions. Therfore all comments are corrected in the manuscript.

Particular comments:

row 62: Tyler instead of „Tylor“

row 70-71: „REE … have higher stability“: correctly perhpas „REE compounds (complexes)…“

row 72-74: note that important REE-bearing rock forming minerals, monazite and xenotime, may also display W-type tetrad effect – rather weak but sufficient to cause M-type tetrad effect in the remaining melt (e.g., Tin and Keppler, 2015; Stepanov et al., 2012).

row 76: „according to one study“ – what study?

row 81: „Y with no 4f electrons is not a pseudo-lanthanide…“ – I think more logical would be: „Y with no 4f electrons IS a pseudo-lanthanide…“ (or behaves like a (pseudo)lanthanide…)

row 88: „rarest elements“ – please specify (I think from the elements mentioned Au, Re and In)

row 126-127: note that Sr, Ba and Zr, usually also Th are rather compatible in peraluminous granites

rows

140-145: note that the granite composition prior to hydrothermal alteration did not have to be identical as the mean upper continental crust. The Figures 2 and 3 are nice but I think that formulations could be modified.

Response: All previous corrections are done in the manuscript.

rows 150-152: is there any evidence (e.g., from relations observed with petrographic microscope or SEM) that the Ti-rich minerals are products of alteration? Note that if Ti-oxides are formed by alteration of biotite (from Ti contained in biotite) it does not enhance Ti content of the rock (or enhances only slightly by passive concentration, if MgO and other compounds are partly removed).

Response Ibrahim et al. (2003) reported the presence of tonalites as older granitoids, from which El Sela granites may be originated.

row 162: regarding Pb, the information about the age of uranium mineralization is important. If it is older than ca. 200-250 millions years, most of the lead would have formed by uranium decay. ).

Response coronadite as Mn-Pb mineral

row 164: micas are usually more important pool of Cs than feldspars (and during alteration, they liberate Cs easily) lepidolite, phlogopite and phlogopite- fulor

(Table 1): very low LOI! (is it from ignition at 1000 or 1100 °C, as usual in igneous rocks, or at lower temperature?) Totals of major component are not shown (or are the values normalized to the sum 100 %?) LOI occurred at1000°C.

rows 182,306,307: „rhenium enhancement…“ – what enhancement? In all analyses shown, Re is below detection limit. For the same reason I have doubts if Re in Figure 3 is correct. Re concentration in ppm, where its natural contets don,t exceed ppb fractions.

row 188: what is the evidence for desilicification? It is difficult to imagine that the rocks contained more than 78 % SiO2 (the actual content) prior to the alteration.

Response Corrected, I agree with you.

rows 190, 192: the cited reference (El Feky et al.) is missing. „residual solutions which are rich in water“ – perhaps better „residual aqueous solutions“?

Response Corrected

rows 198, 201: „dequatzification“ („r“ missing)

Response Corrected

Fig. 4: I do not know what units are used, but in the first chart, the ratio SiO2/Al2O3 is between ca. 2 and 2.5. This contradicts the chemical composition of rocks (from values in Table 1: mass ratio SiO2/Al2O3 is ca. 6, and molar ratio ca. 10, or Si/Al molar ratio ca. 5). Please check the units and calculations in other charts, too.

Response Revised

row 216: note that the pairs La-Nb, La-Ta and Rb-Sr do not represent isovalent elements. The title should be modified.

Response Corrected

row 224, 225: I would not say that deviation of Zr/Hf ratios from the range 33-40 are rare: lower values are common in fractionated magmas. In any case, geochemical or petrological, not technical literature should be cited here.

rows 227,241: „natural“ ratio – replace by „chondritic ratio“ (samples with non-chondritic ratios are natural, too)

Response Corrected

rows 239, 240, 301: is there any evidence of fluorite presence?

Response Petrographically studies were carried out and confirmed the presence of fluorite.

„Chapter 6. Rare Earth Elements“, page 10 (general comment): the major irregularity of the REE distribution is negative cerium anomaly. It is probably combined with a weak M-type tetrad effect forming the first tetrad (and probably the second one – which would be better seen in a chart with the space for promethium, however it is uncertain due to Eu anomaly). Any significant third or fourth tetrad is not observable; in addition, the values t3 and t4 in Table 1 are usually between 0.90 and 1.10, which means that tetrad effect is not significant (t1 cannot be used due to obvious Ce anomaly). The negative Ce anomaly indicates that important portion of trivalent REE (mainly LREE) was brought in an oxidizing solution (depleted in tetravalent cerium which is less soluble), probably together with uranium.

row 266: „Figure 19“??

Response I agree with you.

Figure 6: the chart Ta-U is repeated. Please check the R2 values: I do not believe that R2 = 0,90 in the chart U – Zr/U is correct!

Response Corrected

rows 281,282: the correlation of U and Hf in the chart is weakly negative. Nevertheless, most correlations are insignificant due to low chemical variability of the samples presented. Comparison with little altered or unaltered local granites would be much more interesting.

Response Most samples were collected from U mineralized Box-cut in the shear zone.

row 300: 2x „kasolite“; were the uranium minerals documented?

Response Our mineralogical and the previous studies were added

row 304: „liquids rich phase“?? Later magmatic phases

rows 308-309: the sentence „Increments of gold…“ is repeated (see r. 301-302)

Response Corrected

We thank the Reviewers a lot for the useful and valuable comments that have helped improve the manuscript.

Hoping that all the careful review is sufficient for the direct acceptance of the manuscript, thank you for your time and consideration.

Best wishes,

Author Response

Dear Reviewer,

Please find attached the submission of the carefully revised version of the manuscript in Ref., following the minor comments and modifications of the Reviewers.

Below, the detailed list of the changes made in response to the Reviewer’s minor comments (in italics), outlining every change made a point by point, is provided. The changes are marked in the manuscript text in the colour highlighted for the reviewer.

Comments and Suggestions for Authors

 This manuscript is of some interest, describing the geochemistry of granites with high REE and uranium contents. New geochemical data of altered granite pluton are presented and REE tetrad effect is discussed.

But many questions remain unclear to me.

First of all, this concerns the granitic plutons themselves. It is not clear in what tectonic environment granitic rocks were formed, what is their age, or their mineralogical composition. The geology is practically not described. In Fig 1, is shown ophiolite sequence: serpentinites, meatabasalts, gabbro, etc, but not described. The presence of ophiolites suggests that these are island/continental arc systems (orogenic belts) with suprasubduction ophiolites. There is no age and position of granites relative to ophiolites. Are these granites subduction-related, collisional, or post-orogenic? I understand that fractures (strike-slip faults) are responsible for hydrothermal alteration by fluid-rock interaction. Authors mentioned that the main alteration is kaolinization and hematitization but it is not clear from major element analyses.

I recommend improving geology, characteristics of granite plutons, their position, age, and petrography including photomicrographs to see the hydrothermal alteration of rock-forming minerals and new minerals, e.g. uranium minerals that are mentioned in the conclusion. Response:  Petrographical studies were carried out. Our mineralogical and the previous studies were added

In section “Geological setting” three granite intrusions are mentioned and only one is analyzed. Response: indicated

Granite type (I, S, A, M), ASI, MALI, Fe*, and other characteristics should be described. Table 1 contains 8 altered samples from El Sella granite.

Response: indicated

In the section “Material and Methods” not clear how many samples of two mica and muscovite granite were analyzed.

Response: Two mica granite

 In Fig 4, six samples are shown, on other Figures - 8 to 9 (Fig. 6). Is there unaltered granite for comparison?

Response: added

In Table 1, the total Na2O+ K2O wt. % content is relatively low for muscovite granite. Therefore, unaltered granite content will be important. If not analyzed at least an analysis from the literature should be shown.

Response: added

REE plot should be revised. What chondrite REE value used: Sun and McDonough (1989); Taylor and McLennan (1985), etc. A better use, for example, GCDkit software to show normalized REE patterns.

Response: Rudnick and Gao (2003)

The tetrad effect is characteristic of highly differentiated rare metal granites. Muscovite and two mica granite are mainly related to S-type and are peraluminous but the authors mentioned that granite is metaluminous.

Response: peraluminous

Uranium mineralization should be described in more detail. added

English should be improved ok

Other comments:

Line 18:” 9Institute of Physics and Technology, Ural Federal University, St. Mira, 19, 620002, Yekaterinburg”, - but the co-author's name is missing in Lines 5 and 6  corrected

Line 21: The observed alterations in granitic …” rocks” is missing or should be “granites”  corrected

Line 23: Chemical analyses of samples were carried for major oxides, trace and rare earth elements by ICP- should be “carried out"

Response: corrected

Line 75-76: - Sentence: The occurrence of the tetrad effect in nature has been linked to a history of contact or interaction with water, according to one study. - One study (Reference should be here). It is better to combine with the following sentence and rewrite.

Line 189: Should be “which are”

Line 198: dequatzification should be dequartzification

Fig. 4: In legend analyzed samples are missing

Line 239: In highly differentiated granite Rb/Sr ratio usually high.

Line 247: granitite should be granite

Line 250: Are these granites really metaluminous?

In Fig. 5: in Legend not shown what is 1-8 lines and how normalized.

Line 260: Why Fig 4, maybe Fig 5

Line 266: What is Figure 19?

In Fig. 6: in some graphs are 8 or 9 samples

Line 287-287: Check this sentence

Line 299-301: The presence of uranium minerals is not mentioned and described English should be improved.

Response: the authors accept with your comments and suggestions. Therfore all comments are corrected in the manuscript.

We thank the Reviewers a lot for the useful and valuable comments that have helped improve the manuscript.

Hoping that all the careful review is sufficient for the direct acceptance of the manuscript, thank you for your time and consideration.

Best wishes,

Reviewer 3 Report

Dear all writers,

          This granite is very unusual and the content of manuscript is also interested subject. However, the following questions are not answered in this manuscript.

1) Petrology of this granite is unknown.

2) What does “tetrad” term mean? I honestly do not know this terminology, please use another term.

3) Sample locations are not shown in the geologic map.

4) The relations between silicification zones and fault zones are not shown and not explained in the manuscript. Please give some information about alteration zones, structural geology and regional tectonics to international readers.

5) The objectives of this research are not clearly defined. Why did you do this research? Which problem did you intent to solve in this research? There must be a scientific reason, please define this reason.

6) What is the relation between REE composition and uranium mineralization?

7) What is the relation between REE enrichment and kaolinitization, hematization alteration zones?

Line 102-104: Authors mentioned about ENE-WSW trending quartz veins, but they did not mention about direction of the fault zones. If there is a hydrothermal alteration, there must be more than one fracture zone. Is there any parallel or subparallel relation between quartz veins and fault zones?

Line 111-113: Which international reference materials BIR1, W2, ivIRG-1, JA2 and JG3 did you use for REE analysis in granites? Please give the name of granite standard.

Line 116: It should be Table 1

Fig.2: Series number implies sample numbers in Table 1 and the chemical composition of samples were normalized according to UCC values of Rudnick and Gao (2003).

Line 154: What does Fig. 16 means? You do not have this figure.

Fig.3: normalized according to Rudnick and Gao (2003).

Line 185: El Mezayen et al. (2019) reported ………

Line 198: I have never heard the term of “dequartzification” before?? I believe silica dissolution is a better term. Are you saying quartz dissolution or silica depletion or dissolution?

Line 266. Fig. 19 is not existing in this manuscript.

Line 266: “the fact that Pm does not occur in nature”. What does it mean? Why did you write this sentence?

Line 270: Explain the meaning of T1, T2 and T3.

Line 281: Fig. 5 never shows U mineralization, it should be Fig. 6.

Line 287: The writers reported “Kaolinization, silicification and hematitization are the main alteration processes and associated with uranium and other mineralization” but there is no enough geologic information about these alteration processes and uranium mineralization. Where is the uranium mineralization? So, this conclusion cannot be drawn from this study.

          As conclusion, at the present time, this manuscript is not ready for publication and it requires major revision. Manuscript also requires better English.

Author Response

Dear Reviewer,

Please find attached the submission of the carefully revised version of the manuscript in Ref., following the minor comments and modifications of the Reviewers.

Below, the detailed list of the changes made in response to the Reviewer’s minor comments (in italics), outlining every change made a point by point, is provided. The changes are marked in the manuscript text in the colour highlighted for the reviewer.

Comments and Suggestions for Authors

Dear all writers,

          This granite is very unusual and the content of manuscript is also interested subject. However, the following questions are not answered in this manuscript.

1) Petrology of this granite is unknown.

Response:  Petrographically studies were carried out

2) What does “tetrad” term mean? I honestly do not know this terminology, please use another term. Response: done

3) Sample locations are not shown in the geologic map. .

Most samples were collected from U-mineralized Box-cut in the shear zone.

4) The relations between silicification zones and fault zones are not shown and not explained in the manuscript. Please give some information about alteration zones, structural geology and regional tectonics to international readers.

Response: Added

5) The objectives of this research are not clearly defined. Why did you do this research? Which problem did you intent to solve in this research? There must be a scientific reason, please define this reason.

Response: We want to know, if there is any correlation between uranium mineralization and other mineralization and also determining the prevailed physic-chemical conditions.

6) What is the relation between REE composition and uranium mineralization? Rare earth elements  due to their nearly identical properties are used in illustrating the studied

Response: physic-chemical conditions.

7) What is the relation between REE enrichment and kaolinitization, hematization alteration zones?

Response: Kaolinization as an acidic process must lead REEs and uranium migration, in contrast to hematitization as an alkaline process leads to REEs and uranium accumulation. 

Line 102-104: Authors mentioned about ENE-WSW trending quartz veins, but they did not mention about direction of the fault zones. If there is a hydrothermal alteration, there must be more than one fracture zone. Is there any parallel or subparallel relation between quartz veins and fault zones?

Response: added

Line 111-113: Which international reference materials BIR1, W2, ivIRG-1, JA2 and JG3 did you use for REE analysis in granites? Please give the name of granite standard.

Response: The used references are indicated in the following table 

Line 116: It should be Table 1

Fig.2: Series number implies sample numbers in Table 1 and the chemical composition of samples were normalized according to UCC values of Rudnick and Gao (2003). ok

Line 154: What does Fig. 16 means? You do not have this figure.

corrected

Fig.3: normalized according to Rudnick and Gao (2003).  ok

Line 185: El Mezayen et al. (2019) reported ………

Response: corrected

Line 198: I have never heard the term of “dequartzification” before?? I believe silica dissolution is a better term. Are you saying quartz dissolution or silica depletion or dissolution?

Line 266. Fig. 19 is not existing in this manuscript.

Line 266: “the fact that Pm does not occur in nature”. What does it mean? Why did you write this sentence?

Response: corrected

Line 270: Explain the meaning of T1, T2 and T3.

The term ‘tetrad effect’ in geochemistry refers to the subdivision of the 15 lanthanide elements into four groups in a chondrite normalized distribution pattern: (1) La–Ce–Pr–Nd, (2) Pm–Sm–Eu– Gd, (3) Tb–Dy–Ho, and (4) Er–Tm–Yb–Lu, and each group forms a smooth convex (M-type) or concave (W-type) pattern (Masuda et al., 1987). The values of tetrad effect were calculated according to the quantification method of Irber (1999): t1 = (Ce/Ce*× Pr/ Pr*), t3 = (Tb/Tb*×Dy/Dy*), t4 =(Tm/Tm* ×Yb/Yb*) Degree of the tetrad effect T1,3 = (t1 × t3)0.5. A REE pattern that does not show a tetrad effect has values of TE1.3 < 1.1. The M-shaped pattern shows ТЕi <1.1 and the W-shaped TEi < 0.9. The REE tetrad effect is most visible in late magmatic differentiates with strong hydrothermal interactions or deuteric alteration. This includes highly evolved leucogranites, pegmatites, and mineralized granites. Moreover, the tetrad effect is often accompanied by other modified geochemical behavior of many trace elements, which is termed by Bau (1996) as non-CHARAC behavior (CHARAC =Charge-and-Radius-Controlled). Such behavior occurs typically in highly evolved magmatic systems enriched in H2O, CO2 and elements such as Li, B, F and/or Cl, which suggests the increasing importance of an aqueous-like fluid system during the final stages of granite crystallization (Bau, 1996; Irber, 1999).

Line 281: Fig. 5 never shows U mineralization, it should be Fig. 6.

Line 287: The writers reported “Kaolinization, silicification and hematitization are the main alteration processes and associated with uranium and other mineralization” but there is no enough geologic information about these alteration processes and uranium mineralization. Where is the uranium mineralization? So, this conclusion cannot be drawn from this study.

Response: added

          As conclusion, at the present time, this manuscript is not ready for publication and it requires major revision. Manuscript also requires better English.

Response: revised

We thank the Reviewers a lot for the useful and valuable comments that have helped improve the manuscript.

Hoping that all the careful review is sufficient for the direct acceptance of the manuscript, thank you for your time and consideration.

Best wishes,

Round 2

Reviewer 1 Report

The text has been improved significantly (especially petrographic description has been added), but abstract and conclusions not. Both should be actualized to be consistent with the text. In addition, I think that information that there are some ophiolites in the area (probably unimportant for the mineralization presented) is redundant in the abstract.

The chapter "4. Alteration processes in the studied altered granites " remains inconsistent with data. As evident from comparison of mineralized granites with unaltered ones (according to completed Table 1), the mineralized ones are enriched in Si, depleted in K and strongly depleted in Na. Thus, the bulk chemical composition brings no evidence for processes like K-metasomatism or even dequartzification in the mineralized granites (in contrast, they may have occurred in their surroundings - but this is only a speculation).

Also the problem with rhenium remains: only one of 8 analyses in the revised Table 1 shows Re concentration equal to detection limit, i.e. 2 ppb, all others are lower. Thus, the mean Re concentration in the mineralized granites is probably similar to the mean crustal value of 1 ppb which you stated (a reference could be added). It is surely not ca. 10 times higher than crustal average, as displayed in Figure 4!

Another problem is with tantalum: Ta is not shown in Table 1, but the comparison of La/Nb and La/Ta ratios implies that the Nb/Ta ratio is close to 2 (or the La/Ta ratios must be much higher than what is written in the text - see rows 282-284). Note that low (rather than high) Nb/Ta ratio is consequence of magmatic differentiation.

Regarding the tetrad effect: from the data presented, there is only one statistically significant value which could mean presence of the fourth tetrad (t4 = 1.12). All t3 values are lower than 1, implying that there is no third tetrad (and if so, it is of a weak W-type; however this is improbable). Presence of M-type in first two tetrads (combined with Ce- and Eu anomalies) is possible (I think that even probable, however, the accuracy of values of all La, Pr, Nd, Sm and Gd is crucial).

Chapter "Geological setting" has been improved but orientation is still somewhat difficult. Please, split the text into more paragraphs!

As stated in the rows 110-111: "A microgranite dike hosts the most radioactive anomalies in the study area". I suppose that this dyke does not represent the altered and mineralized granites presented in the manuscript, as their grain size stated (3-3.5 mm) does not correspond to microgranite. However, its relation to the altered granites should be clarified. Is this vein yet more enriched in uranium? Do the altered granites presented represent the whole shear zone or some specific domains?

Other comments:

row 129: what is "flogopite - fulor"?

rows 176, 177: please note that Zr, Th, Sr and Ba are compatible elements in peraluminous granites. Also note that you are writing that zircon probably crystallized early (rows 222-223), and that feldspar crystallization decreases Sr content of residual magma (rows 285-286) - I agree with that.

rows 224-225: "kasolite" 2x

row 289: replace "natural ratio" by "chondritic ratio" (or "crustal ratio"?)

rows 326-327: "weak positive correlation..." OF WHAT? "...with Hf, Ag and Zr/U"

row 346: "kasolite" 2x again

rows 353-354: "abundance of Mo for rhenium..." - what? Perhaps you mean molybdenite?

rows 358-359: the correlation of U with the Zr/U ratio is (logically) negative, as also evident from the chart (Fig. 7f)

Author Response

Dear Reviewer,

Please find attached the submission of the carefully revised version of the manuscript in Ref., following the comments and modification of the reviewer.

Response: The requirements were done in the manuscript.

We thank a lot the Reviewer for the useful and valuable comments that have helped to improve the manuscript.

Hoping that all the careful review is sufficient for the direct review of the manuscript, thank you for your time and consideration.

Best wishes,

Reviewer 2 Report

The authors have made appropriate additions, but I still have no clarity regarding granites. Silica alteration is not visible on granite microphotographs, and in some photos the alteration is insignificant. Authors mentioned that granites belong to the A-type, however, the amount of alkalis is low, and unaltered granite looks alien in relation to the studied granites, low in phosphorus, while altered ones are rich in phosphorus. In the first review, I recommended including fresh granite analyses, now I recommend deleting it

On the REE diagram, it is necessary to indicate whose data were used in normalization.

Author Response

(The authors gave the same response as above.)
